# Involvement of CB2 Receptors in the Neurobehavioral Effects of *Catha Edulis* (Vahl) Endl. (Khat) in Mice

**DOI:** 10.3390/molecules24173164

**Published:** 2019-08-30

**Authors:** Berhanu Geresu, Ana Canseco-Alba, Branden Sanabria, Zhicheng Lin, Qing-Rong Liu, Emmanuel S. Onaivi, Ephrem Engidawork

**Affiliations:** 1Department of Pharmacology and Clinical Pharmacy, School of Pharmacy, Addis Ababa University, 1176 Addis Ababa, Ethiopia; 2Department of Biology, William Paterson University, Wayne, NJ 07470, USA; 3Department of Psychiatry, Harvard Medical School, Psychiatric Neurogenomics, Division of Alcohol and Drug Abuse, and Mailman Neuroscience Research Center, McLean Hospital, Belmont, MA 02478, USA; 4Laboratory of Clinical Investigation, National Institute on Aging, National Institutes of Health, Baltimore, MD 21224, USA

**Keywords:** khat, endocannabinoid system, tyrosine hydroxylase, dopamine transporter, locomotor activity, MPTP, JWH133

## Abstract

There is behavioral evidence for the interaction between crude khat extract and the endocannabinoid system, whereby the endocannabinoid system alters khat extract-mediated behavioral effects through modulation of the monoaminergic system. The objective of this study was to investigate the role of the endocannabinoid system on the neurobehavioral effect of khat extract in mice following concomitant administration of khat extract and the CB2R agonist, JWH133. Locomotor activity test, immunohistochemistry, and reverse transcriptase polymerase chain reaction technique were utilized to assess locomotor activity, tyrosine hydroxylase immunoreactivity, and expression of dopamine transporter mRNA gene. The results show sub-acute administration of khat extract alone increased locomotor activity in mice and co-administration of the CB2R agonist, JWH133, reduced khat extract induced hyperlocomotor activity. The data revealed that cell type specific deletion of CB2Rs on dopaminergic neurons increased the hyperlocomotor behavior of khat extract. Furthermore, the results revealed that khat extract attenuated MPTP induced motor deficits, which is enhanced by JWH133. Khat extract also increased expression of tyrosine hydroxylase positive cells and expression of dopamine transporter mRNA gene in wild type mice. Nevertheless, JWH133 did not alter the effect of khat extract on tyrosine hydroxylase immunoreactivity and dopamine transporter mRNA expression when given together with khat extract. Taken together, the results suggest that the CB2Rs selectively interact with khat extract-mediated locomotor effects and could be utilized as therapeutic target in central nervous system movement disorders associated with dopamine dysregulation.

## 1. Introduction

Khat, *Catha edulis* (Vahl) Endl., has been consumed for centuries by people living around the horn of Africa and the Middle East [1,2]. Cathinone is the main active constituent of khat leaves and has the same mode of action as amphetamine [3], and behavioral effect of cathinone in animals is similar to amphetamine [3,4,5]. Khat chewing results in different CNS (central nervous system) effects including euphoria, excitation, anorexia, increased respiration, hyperthermia, analgesia, increased sensory stimulation [5,6], increased locomotion, and altered performance in several behavioral experiments in rodents [7,8,9].

The endocannabinoid system (ECS) is an endogenous neuromodulatory system composed of the two major cannabinoid receptors (CBRs, CB1Rs, and CB2Rs), endogenous ligands or the endocannabinoids (eCBs) and enzymes responsible for the synthesis and degradation of eCBs [10]. The ECS modulates numerous central nervous system (CNS) functions, including the classic cannabinoid tetrad of thermoregulation, antinociception, locomotor activity, and catalepsy [11], as well as feeding behavior [12], food reinforcement [13], and cognition [14]. CB1Rs have been well characterized using conditional *Cnr1* mutant mice [15,16,17,18] However, CB2Rs were previously thought to be in immune cells, were referred to as peripheral CB2Rs, and were less investigated for neuronal CNS function [10]. Now conditional *Cnr2* mutant mice have been generated using the Cre-LoxP technology to produce Syn-*Cnr2* conditional knockout (cKO) mice in which synaptic deletion of CB2Rs was shown to mediate a cell type specific plasticity in the hippocampus [19]. A similar approach was used to produce DAT-*Cnr2* cKO mice with specific deletion of CB2Rs from dopamine (DA) neurons [20]. Thus, contrary to previous thoughts [21], there is now overwhelming evidence for the functional neuronal expression of CB2Rs and their presence detected in the hippocampus, striatum and brain stem [19,20,22,23,24,25,26]. CB2Rs expressed in mouse brain DA neurons are involved in drug reward; synaptic plasticity; drug addiction; eating disorders; and psychosis, depression, and autism spectrum disorders. Recently, Liu et al. [20] showed that CB2Rs in DA neurons are involved in motor function, and their deletion releases the “brake” on psychomotor activity and results in continuous spontaneous hyperactivity. Hence, DA neuron cell type-specific CB2R cKO mice are relevant in studying the molecular basis of CB2R neuronal signaling mechanism and their role in different behavioral modifications.

Studies showed that the dopaminergic pathway is involved in the behavioral effects observed by the activation of ECS [27,28] and by administration of khat extract in animals [29,30]. Major dopaminergic systems in the brain originate from brainstem DA neurons located in the substantia nigra pars compacta (SNc) and the ventral tegmental area (VTA). Dopamine is synthesized from tyrosine by the rate-limiting enzyme tyrosine hydroxylase (TH), accumulated into synaptic vesicles by the vesicular monoamine transporter-2 (VMAT2), and its effect is terminated mainly by re-uptake into dopaminergic terminals by the dopamine transporter (DAT) [31,32,33]. It is well established that DA neurotransmission in both dorsal and ventral striatum is essential for normal locomotor functions, and progressive degeneration of DA neurons in these areas is a known cause of Parkinson disease (PD). 1-Methyl-4-phenyl-1,2,3,6-tetrahydropyridine (MPTP) is commonly used to induce dopaminergic toxicity to produce neuropathologic abnormalities resembling the idiopathic PD in humans [34].

Several lines of experimental evidence on wild type (WT) and cannabinoid receptor manipulated animals have shown the role of the ECS on the effect of psychoactive substances, including opioids [35], nicotine [36,37], and cocaine [38]. There is also evidence for the involvement of the ECS in the behavioral effects of khat extract in animals [8]. Since the mesocorticolimbic dopaminergic pathway plays a vital role in mediating some of the behavioral effects of both khat extract and cannabinoids, studying the interaction of the ECS and khat extract will provide an insight into the identification of drug targets and the development of new pharmacologic approaches for the treatment of CNS disorders associated with dopamine dysregulation. To this effect, the involvement of the ECS on the neurobehavioral effect of khat extract was investigated in DAT-*Cnr2* cKO mice and WT mice following concomitant sub-acute administration of khat extract and the CB2R agonist, JWH133. In addition, the behavioral effect of sub-acute exposure of khat extract was investigated in mice treated with MPTP.

## 2. Results

### 2.1. Khat Extract Increases Locomotor Activity in WT Mice

Total distance travelled and stereotypic counts are among the most commonly used parameters to test mice motor behavior in the activity monitor apparatus. An independent *t*-test showed that administration of khat extract at 300 mg/kg to the WT mice significantly (*p* < 0.001) increased both the total distance travelled by 36.2% and stereotypic count by 32.7% in the activity box compared to controls (Figure 1 and Figure 2).

### 2.2. Deletion of CB2Rs in DA Neurons Augments Locomotor Activity of Khat Extract

CB2Rs were previously thought to be predominantly expressed in immune cells and their involvement in cannabinoid-induced behaviors was largely unexplored. Recent studies showed that CB2Rs inhibits motor function and the deletion of CB2Rs in DA neurons induces enhanced motor function characterized by hyper-locomotion of the DAT-*Cnr2* cKO mice. We evaluated the hyper-locomotor activity of DAT-*Cnr2* mice and the role of deletion of CB2Rs in DA neurons on the locomotor activity of khat extract. The result from *t*-test depicts that deletion of CB2Rs in DA neurons per se significantly (*p* < 0.001) increased distance travelled in the activity box by 32% and stereotypic count by 28%. Administration of khat extract in a dose of 300 mg/kg significantly (*p* < 0.001) increased both the total distance travelled by 20.8% and stereotypic count by 34.9% in the DAT-*Cnr2* mice compared to the WT (Figure 1 and Figure 2).

### 2.3. Effect of JWH133 on Khat Extract′s Locomotor Activity

The effect of co-administration of the selective CB2R agonist JWH133 on the locomotor activity of khat extract in WT mice was evaluated using the activity monitor apparatus. One-way ANOVA showed there was significant association between distance travelled (F(3,20) = 14.509, *p* < 0.001) and treatment groups. Tukey′s post hoc test showed that JWH133 significantly (*p* < 0.05) attenuated khat extract induced increase in distance travelled by 19.1% in the activity box. There is also a statistically significant difference (F(3,20) = 13.285, *p* < 0.001) in stereotypic count among the treatment groups. In a post hoc analysis, JWH133 significantly (*p* < 0.01) reduced the effect of khat extract by 25.5% on stereotypic count. On the other hand, JWH133 alone tended to reduce the distance travelled and stereotypic count compared to the control group, but the reduction was not statistically significant. In addition co-administration of JWH133 did not show apparent difference in distance travelled and stereotypic count compared to the control group (Figure 3A,B).

### 2.4. Khat Extract Attenuates MPTP-Induced Motor Deficits

The effect of khat extract on locomotor activity was evaluated in sub-acute MPTP lesioned mice. The locomotor activity was reduced in MPTP group compared to that of control group, confirming that the PD model was successfully established. Moreover, the locomotor activity of khat extract treated group was increased compared with that of MPTP group, which indicated that khat extract attenuated the motor deficits of PD mouse. One-way ANOVA showed that there is a statistically significant difference in distance travelled among the treatment groups (F(4,25) = 54.029, *p* < 0.001). Post hoc test showed that PD model mice treated with MPTP had a significant (*p* <0.001) reduction in distance travelled by 52.9% compared to the control group, and this was significantly (*p* <0.01) attenuated by 59.1% following treatment with khat extract in PD mice (Figure 4A). There is also a statistically significant difference (F(4,25) = 52.921, *p* < 0.001) in stereotypic count among the treatment groups. Mice in the MPTP group had a reduced (*p* < 0.001) stereotypic count by 46.7% compared to those of the control group, which was reversed (*p* < 0.01) by 48.6% after khat extract treatment (Figure 4B).

### 2.5. JWH133 Increases Hyperlocomotor Effect of Khat Extract in MPTP Lesioned Mice

The effect of administration of JWH133 alone and its co-administration on the locomotor activity of khat extract was also evaluated in sub-acute MPTP lesioned mice. One-way ANOVA showed that there is a statistically significant difference in distance travelled (F(4,25) = 54.029, *p* < 0.001) and stereotypic count (F(4,25) = 52.921, *p* < 0.001) among the treatment groups. Administration of JWH133 alone significantly (*p* < 0.01) reversed MPTP induced reduction in total distance travelled by 36% and stereotypic count by 28%. JWH133 also significantly (*p* < 0.001) increased khat extract induced increase in distance travelled by 15.9% (Figure 4A) and stereotypic count by 7.6% (Figure 4B) compared to MPTP lesioned mice in the activity box in post hoc test.

### 2.6. Effect of Khat Extract on Immunohistochemical Staining for TH-Positive Neurons in WT Mice

To examine the intactness of dopaminergic neurons, brain sections were immunostained for TH immunoreactivity in the VTA and SNc (Figure 5A) of the WT mice. The results of one-way ANOVA revealed that there is significant difference (F(3,8) = 6.554, *p* < 0.01) among groups in TH immunoreactivity in the VTA, but no apparent difference was detected in the SNc region. Tukey′s test showed that mice treated with khat extract had significantly (*p* < 0.05) increased (12%) TH immunoreactive cells in the VTA region, compared to controls (Figure 5B).

### 2.7. JWH133 Did Not Alter TH Immunoreactivity of Khat Extract

Co-administration of JWH133 with khat extract (300 mg/kg) did not significantly alter TH immunoreactivity of khat extract either in the VTA or SNc region compared to khat extract alone treated mice (Figure 5A,B).

### 2.8. Effect of Khat Extract on DAT Mrna Gene Expression in WT Mice

The effect of khat extract on the expression of DAT mRNA was investigated using qRT-PCR technique. qRT-PCR was used to estimate DAT mRNA in the VTA region of WT mice. The study demonstrated that khat extract administration increased the expression of DAT mRNA twofold compared to vehicle treated controls. To test for statistically significant difference, one-way ANOVA was done. The result revealed that there is statistical difference in DAT mRNA expression among the groups (F(3,8) = 60.488, *p* < 0.001), and post hoc test showed that khat extract significantly (*p* < 0.001) increases expression of DAT mRNA compared to the control (Figure 6).

### 2.9. Effect of Co-Administration of JWH133 on DAT Mrna Gene Expression

There was no apparent difference detected in the expression of DAT mRNA in the VTA region of mice treated with a combination of JWH133 and khat extract compared to mice treated with khat extract alone (Figure 6).

## 3. Discussion

A large body of evidence exists in the literature, since its first description by Peter Forskal [39], about the adverse health aspects of khat, with a few mentions of its pharmacology [3,40,41,42,43]. However, there are little or no studies that attempted to address the endogenous systems involved and the molecular mechanism for the observed effects of khat extract alone or its combination with other compounds. The aim of the present study was therefore to investigate the role of the ECS on the neurobehavioral effects of khat extract in wild type and MPTP lesioned mice following concomitant administration of khat extract and the CB2R agonist, JWH133. Furthermore, mice with cell type specific deletion of type 2 cannabinoid receptors were used to probe involvement of the ECS on the neurobehavioral effects of Khat extract.

The result of this study showed that sub-acute administration of crude khat extract significantly increased locomotor activity in WT mice, which is in line with previous studies [7,44,45]. The stimulatory effect of khat extract is perceived as an increase in alertness and energy as well as relief from fatigue. Indeed, these effects have been reproduced in rats after oral administration of different concentrations of khat extract in which higher doses of the plant increased motor activity [46]. However, administration of the CB2R agonist, JWH133, with khat extract in WT mice abated khat extract-induced increase in locomotor activity. The agonist did not affect performance of vehicle-treated rats, although it reduced the performance of khat-treated rats and brought down the level to that of the vehicle-treated rats, suggesting that the cannabinoid system antagonizes the khat mediated system in this paradigm. By contrast, cell type specific deletion of CB2R from dopaminergic neurons alone or administration of khat extract in these animals significantly increased the distance travelled and the stereotypic count.

Synthetic, plant-derived, and eCBs have powerful effects on motor activity in animals which are mediated by CB1 receptors in the brain [47,48]. The presence of functional cannabinoid CB2 receptors in the brain has been controversial, and it is generally believed that the behavioral and psychotropic effects of cannabinoids are mediated by CB1 receptors and CB2 receptor ligands have no psychoactive effects. However, the purported lack of brain CB2 receptors has been challenged by recent reports of CB2 receptors on microglia [49] and neuronal cells [26,50,51] in several brain regions including the anterior olfactory nucleus, cerebral cortex, cerebellum, hippocampus, striatum, and brainstem. Furthermore, activation of CB2 receptors by 2-arachidonoylglycerol, JWH015, or JWH133 inhibits locomotion, morphine-6-glucuronide-induced emesis, and neuropathic pain, while stimulating neural progenitor proliferation and producing neuroprotective effects [52,53,54].

The observation that deletion of CB2Rs in DA neurons resulted in enhanced spontaneous motor activity, which is in agreement with previous studies [20,23], reinforces the notion that CB2R mediates inhibition of spontaneous movement via modulation of the dopamine system, probably through reduction of neuronal firing frequency [55]. Further evidence for the inhibitory role of CB2R comes from the observation that khat extract′s effect on locomotion was attenuated when the receptor is activated but accentuated in the absence of the receptor. This finding reaffirms our previous observation [8] and strongly suggests that there is an interaction of khat extract and the ECS at the level of the dopaminergic system to modulate motor behavior that could probably have relevance to dopamine-related CNS disorders. The fact that khat extract results in hyperlocomotor behavior in mice lacking CB2Rs in DA neurons provides evidence that brain CB2Rs may constitute a new therapeutic target for treatment of such disorders.

In this study effect of the ECS and khat extract on nigrostriatal DA neurons degeneration induced by MPTP was also investigated by using MPTP lesioned mice. MPTP is a commonly used neurotoxin to induce a PD-like state in animals. As animals do not develop PD, the MPTP lesion model is one of several models that are used to investigate the underlying mechanisms of PD and to test novel compounds for their neuroprotective properties [56,57]. The result of the present study showed that MPTP reduced locomotor activity in mice and khat extract attenuated MPTP-induced motor deficits in mice. In addition, the result demonstrated that administration of JWH133 alone or in combination with khat extract showed an anti-parkinsonian and likely a neuroprotective effect as evidenced by increased locomotor activity in MPTP lesioned mice after co-administration of khat extract and JWH133. Previous findings have shown that activation of the CB2Rs rescued nigrostriatal DA neurons from MPTP neurotoxicity [58,59,60,61], but the effect of co-administration of khat extract and JWH133 on locomotor activity in MPTP lesioned mice had not been investigated. The result suggests that khat extract alone or in combination with the CB2R agonist might have a neuroprotective effect and could probably have a significant role in reversing the motor deficits observed in PD patients.

In the current study immunohistochemical staining for TH positive cells revealed that mice treated with khat extract showed a significant increase in TH immunoreactivity in the VTA compared to the controls but JWH133 did not alter TH expression in mice treated with the drug, which is in agreement with previous study [62]. The result also showed that there was no apparent difference detected in TH immunoreactivity in the SNc region of khat extract treated mice compared to controls. Although TH-immunoreactivity was not performed in MPTP-lesioned mice, the fact that increased immunoreactivity was observed in VTA but not in SNc in normal mice suggests that khat extract may exert region specific effect during physiological and pathological conditions. Evidence for this assertion comes from the observation that though dopamine neurons in both regions share many physiological properties [63,64,65], they markedly differ in their sensitivity to addictive drugs such as nicotine [66].

Tyrosine hydroxylase is the rate-limiting enzyme of catecholamine biosynthesis [67]. The activity of the enzyme is regulated by two mechanisms; TH protein synthesis and phosphorylation. Catecholamines bind the active site of the enzyme and produce feedback inhibition but phosphorylation substantially decreases affinity of catecholamines thereby relieving feedback inhibition and increasing enzyme activity [68]. Since cathinone, the active principle of khat, has similar mechanism of action [3] and related structure [69] to amphetamine, it is worth expecting that khat extract has similar dopamine releasing effect to amphetamine. In this context the increased synaptic dopamine levels after khat are subject to enzymatic degradation rather than repacking into vesicles like the case of amphetamine [70], which ultimately results in increased dopamine turnover. So the increase in TH-immunoreactivity after khat extract administration might be attributed to an enhanced dopamine turnover and consequently an enhanced expression of the enzyme required for its biosynthesis. Taken together, enhanced locomotor activity and increased expression of TH-reactive cells after khat extract administration may suggest that khat extract could have a significant therapeutic effect in PD. There was enhanced TH immunoreactivity and reduced DAT gene expression in the frontal cortex and VTA of mice with selected deletion of type 2 cannabinoid receptors in dopamine neurons compared with the WT mice as we reported [23,71].

However there is a need for further studies to establish the molecular mechanisms involved in anti-PD potential of khat extract. However, the adverse effects of khat extract including disturbance in mood, hallucinations, delusions, elevation of arterial blood pressure, risk of carcinoma of the mouth and esophagus, anorexia, constipation [5,6] may hamper its therapeutic use for long lasting or chronic movement disorders. Although khat is known to have several limitations to be used for therapeutic purposes, future research should address whether khat or the sprayed pesticides are the culprit for most of the side effects as well as how to minimize intrinsic adverse effects associated with khat use.

The effect of khat extract administration on the expression of DAT mRNA was investigated in this study using qRT-PCR technique in the WT mice. The result demonstrated that khat extract administration resulted in a significantly increased expression of DAT mRNA by two-fold compared to vehicle treated controls. However, there was no apparent change in the expression of DAT mRNA when khat extract was administered in combination with JWH133 compared to khat extract alone treated mice. Dopamine transporter is a Na^+^/Cl^−^ dependent transporter responsible for the uptake of extracellular DA into the nerve terminal and hence terminating the effect of DA [72].

Previous studies on the effect of amphetamine on DAT expression showed that administration of amphetamine increases surface expression of DAT [72,73,74]. The relationship between DAT expression and amphetamine may be due to several factors. First, higher DAT levels likely result in increased amphetamine transport and thus augmented intracellular amphetamine concentrations. In addition to increased cytoplasmic amphetamine levels, higher DAT levels may also allow for augmented DAT coupling to VMAT-2 on synaptic vesicles, which together produce greatly augmented intravesicular amphetamine concentrations and thereby greater vesicular depletion via amphetamine-induced disruption of vesicular pH gradients. Increased vesicular depletion results in augmented cytoplasmic dopamine levels, which can then be moved into the extracellular space via DAT-mediated reverse transport more readily due to augmented DAT levels [70,75]. Hence, the effect of khat extract on DAT mRNA expression in this study might be viewed similarly to the effect of amphetamine since the two compounds are presumed to have similar effect as mentioned above. Dopamine modulates physiological processes like locomotor activity, cognitive processes, reward, and addiction. Dysfunction of the dopaminergic system is thought to be attributable to the development of multiple neurological disorders such as schizophrenia, PD, depression, attention deficit hyperactivity disorder (ADHD) and drug addiction [76]. This suggests that Khat modulation of dopaminergic system could be exploited in studies of CNS disorders associated with dopamine dysfunction.

## 4. Materials and Methods

### 4.1. Animals

The experiments were performed using DAT-*Cnr2* cKO and C57BL/6J WT mice. The generation of DAT-*Cnr2* cKO mice, genotyping and RNAscope in situ hybridization has been described elsewhere [20]. Briefly, we generated *Cnr2*-floxed mice that were crossed with DAT-*Cre* mice, in which *Cre* recombinase expression is under DAT promoter control to ablate *Cnr2* gene in midbrain DA neurons of DAT-*Cnr*2 cKO mice. The experiments were performed in adult mice (20–30 g body weight) and were bred in the mouse laboratory at William Paterson University. The animals were maintained under controlled room temperature (25 ± 2 °C) and light-dark (12:12 h) conditions with free access to food and water. The experimental procedures followed the Guide for the Care and Use of Laboratory Animals and were approved by William Paterson University animal care and use committee (IACUC).

### 4.2. Drugs and Chemicals

The CB2R agonist, JWH133 was purchased from Sigma-Aldrich Chem. Co. (St. Louis, MO, USA) and MPTP was purchased from Cayman Chem. Co. (Ann Arbor, MI, USA).

### 4.3. Collection and Extraction of Khat

Fresh khat leaves were purchased from a local market in Aweday, 515 km east of Addis Ababa, Ethiopia, which is one of the common natural habitats. The collection of the plant was done at the same time and from same location. The fresh bundles were packed in plastic bags and transported in an ice box to the Department of Pharmacology and Clinical Pharmacy, School of Pharmacy, Addis Ababa University. The fresh leaves were then immediately kept in a deep freezer (−20 °C).

Khat extraction was performed as described elsewhere [77]. Briefly, the leaves were finely chopped, weighed by electronic digital balance and placed in an Erlenmeyer flask containing a mxture of diethyl ether and chloroform in a 3:1 ratio. Enough volume of the solvent was added so as to cover the crushed plant material in the flask. The flask was wrapped with aluminum foil and the contents were continuously stirred using a rotary shaker (New Brunswick Scientific Co, CT, USA) at 120 rpm for 24 h at room temperature. It was then filtered using Whatman No.1 filter paper (90 mm diameter, Whatman Ltd., Maidstone, England) and concentrated in a hood for 24 h. The concentrated extract was then poured on a flat container and subjected to freeze drying using a lyophilizer (OPERAN Lyophilizer, Gyeonggi-do, KOREA). The percentage yield (calculated using the weight of fresh khat leaves and weight of the extract) was found to be 1% and the extract was kept in a tightly sealed container in a deep freezer at −20 °C until use.

### 4.4. Grouping and Dosing of Animals

The C57BL/6J WT mice were grouped (n = 6 per group) into CTR (control), JWH133, KHAT and JWH133 + KHAT groups, whereas the DAT-*Cnr2* cKO mice were grouped (n = 6 per group) into CTR and KHAT group. The controls received a vehicle composed of a mixture of Tween, DMSO, and saline solution (1:2:7); mice in the JWH133 group received JWH133 (5 mg/kg); the KHAT group received khat extract (300 mg/kg) alone; the JWH133 + KHAT group received a combination of JWH133 (5 mg/kg) and khat extract (300 mg/kg) where-ever applicable. The various doses were selected based on previous reports [20,45]. The use of a single dose of khat extract was based on previous studies [45] in which khat extract produced a better pharmacologic effect at a dose of 300 mg/kg than other doses. Khat sample containers including syringes were covered with aluminum foil to avoid light decomposition. Khat extract and JWH133 were made up fresh in the vehicle to a predetermined concentration and administered intraperitoneally (i.p.) in a volume of 10 mL/kg body weight for seven consecutive days. The vehicle was given to the control animals in the same volume for same days as khat extract or JWH133.

### 4.5. MPTP-Induced Dopaminergic Lesion in Mice

The C57BL/6J WT mice were further grouped (n = 6 per group) into MPTP, KHAT + MPTP, and JWH133 + KHAT + MPTP groups. Animals in the MPTP group were injected with MPTP 25 mg/kg once in a day for seven consecutive days to induce sub-acute parkinsonian symptoms in mice [78]. Mice in KHAT + MPTP group were injected with khat extract 300 mg/kg in combination with MPTP 25 mg/kg. The JWH133 + KHAT + MPTP group were injected with JWH133 5 mg/kg, khat extract 300 mg/kg and MPTP 25 mg/kg. JWH133 and khat extract were administered 45 min prior to MPTP administration. All injections were through the i.p. route once daily for seven consecutive days.

### 4.6. Spontaneous Locomotor Activity Test

Spontaneous locomotor activity was evaluated by using an infrared photobeam controlled open-field test chamber (MED Associates Inc., St. Albans, VT, USA). Mice were individually placed in the center of the box (43.2 × 43.2 × 30.5 cm) and allowed to freely explore the chamber for 10 min. The test boxes were connected to a computer and total distances travelled (ambulatory distance travelled in centimeters) and stereotypic count (stereotypic distance travelled in beams in the activity monitor) were obtained after 45 min of khat extract or vehicle administration. In order to avoid any clues, the activity cages were cleaned thoroughly with alcohol after each test. All measurements were performed under normal lighting in a ventilated and quiet room.

### 4.7. Immunohistochemical Staining for TH-Positive Neurons

Mice were intracardially perfused with 0.9% saline followed by 4% paraformaldehyde (PFA). The brains were collected and postfixed overnight at 4 °C, cryoprotected with 30% sucrose in phosphate buffured saline (PBS), and then frozen and stored at −80 °C until analysed. Coronal sections (30 μm) containing the SNc and VTA were obtained. The slices were incubated overnight at 4 °C with polyclonal rabbit anti-TH antibody (Abcam, Cambridge, MA, USA). After washing, sections were incubated with goat polyclonal anti-rabbit secondary antibody, Alexa Flour^®^ 488 (Abcam, Cambridge, MA, USA). The sections were mounted on slides and examined using confocal microscopy. Quantification of TH immunostaining was carried out on high-resolution digital microphotographs taken with a 20× objective and under the same conditions of light and brightness/contrast. Slides were used to measure the mean density of labeling in the selected area, using the analysis software ImageJ (Wayne Rasband, NIH, Bethesda, MD, USA), which allows calibration that minimizes the influence of different backgrounds.

### 4.8. Quantitative Reverse-Transcription (Qrt) PCR for Mrna Quantification

We used qRT-PCR to estimate the expression of DAT mRNA in mouse midbrain. After designing two pairs of intron-spanning PCR primers with a temperature of 56–60 °C, one of them was selected for qRT-PCR based on a single peak in melting curve, an amplification coefficient (AC) of “2” in a series of dilutions assay and/or a lower Ct value.

### 4.9. Sampling of VTA Tissue from Mice Midbrain

Adult mice were sacrificed by rapid decapitation for brain collection. Midbrains were promptly dissected in metal mouse matrix (ZIVIC, PA, USA) from the brains, and the VTA was sliced out of the dissected coronal sections and transferred to pre-chilled tubes for RNA extraction.

### 4.10. cDNA Synthesis

Using Verso cDNA synthesis kit (ThermoFisher Scientific) 100 ng RNA was reverse-transcribed into cDNA with oligo dT primers according to the manufacturer′s guidelines. cDNA was diluted by 5 folds with DNase-free water prior to quantification by qRT-PCR or before being stored at −20 °C.

### 4.11. qRT-PCR Analysis of Relative mRNA Levels

We used the Bio-Rad CFX Connect real-time system (Bio-rad, CA, USA) to amplify cDNA samples in triplicates or quadruplicates by incubation. The cDNA samples were amplified at 95 °C for 5 min, then for 49 cycles of 95 °C for 15 sec, 55 °C for 20 sec, and 72 °C for 30 sec using SsoAdvanced Universal SYBR green supermix (Bio-rad, CA, USA) in a final volume of 12.5 μL, with 1 μL of cDNA and a final concentration of 0.5 μM for forward and reverse primers. Amplification coefficient (AC) was calculated from the Ct slope of the standard curve using the following formula: AC = 10^−1/slope^. Serial dilutions (1:2) of the first template were set to generate eight points, and the Ct vs log cDNA concentration plot was constructed to calculate the Ct slope. mRNA level was determined using the AC value, which were normalized with a reference gene, glyceraldehyde-3-phosphate dehydrogenase (GAPDH).

### 4.12. Statistical Analysis

All data were expressed as mean ± standard error of the mean (SEM). Sigma Plot 12.0 statistical program was used. The statistical analysis was performed by student *t*-test and one-way analysis of variance (ANOVA). Post hoc comparisons of means were carried out with Tukey′s test for multiple comparisons when appropriate. Values were considered statistically significant at *p* < 0.05. We used *t*-test to analyze the effect of khat extract on locomotor activity in wild and DAT-*Cnr2* mice. One-way ANOVA together with post hoc Tukey′s test was used to see the effect of JWH133 on khat extract′s locomotor activity, effect of khat extract on MPTP induced motor defficts, effect of khat extract on immunohistochemical staining for TH positive cell, and effect of khat extract on the expression of DAT mRNA.

## 5. Conclusions

In conclusion, the seemingly opposite effect of khat extract and the CB2 agonist suggests that the CB2R activation has a negative effect on the positive modulatory effect of khat extract on dopaminergic-mediated movement. Khat extract attenuated MPTP-induced motor deficit and increased TH-immunoreactivity and DAT mRNA expression. These effects, however, were not enhanced by concurrent administration with the CB2 agonist, suggesting that the CB2R selectively interact with khat extract-mediated neurobehavioral effects. The results of the study may contribute to the way forward in the search of new therapeutic targets and therapeutic options in the treatment of CNS disorders associated with dopamine dysregulation. However future studies should be done to further our understanding of the detailed molecular mechanisms actions of khat extract.

## Figures and Tables

**Figure 1 molecules-24-03164-f001:**
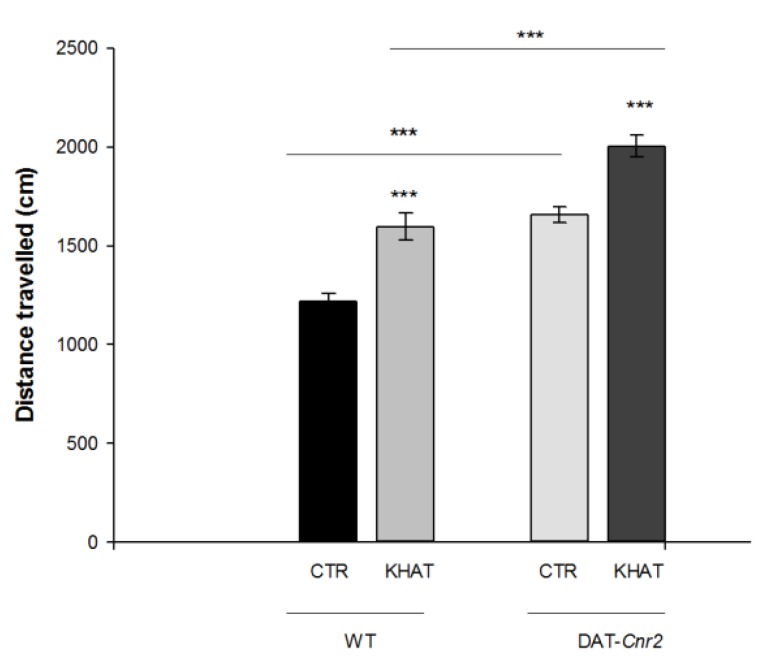
Sub-acute effect of khat extract on distance travelled (centimeters) in the activity monitor apparatus in wild type (WT) and DAT-*Cnr2* mice. Values are mean ± SEM (n = 6 in each group). Statistical analysis was done using student *t*-test. *** *p* < 0.001. CTR (control group received the vehicle), KHAT (treatment group received khat extract in a dose of 300 mg/kg). CTR (control group received the vehicle), KHAT (treatment group received khat extract in a dose of 300 mg/kg). The vehicle and khat extract were administered to mice once daily for seven consecutive days.

**Figure 2 molecules-24-03164-f002:**
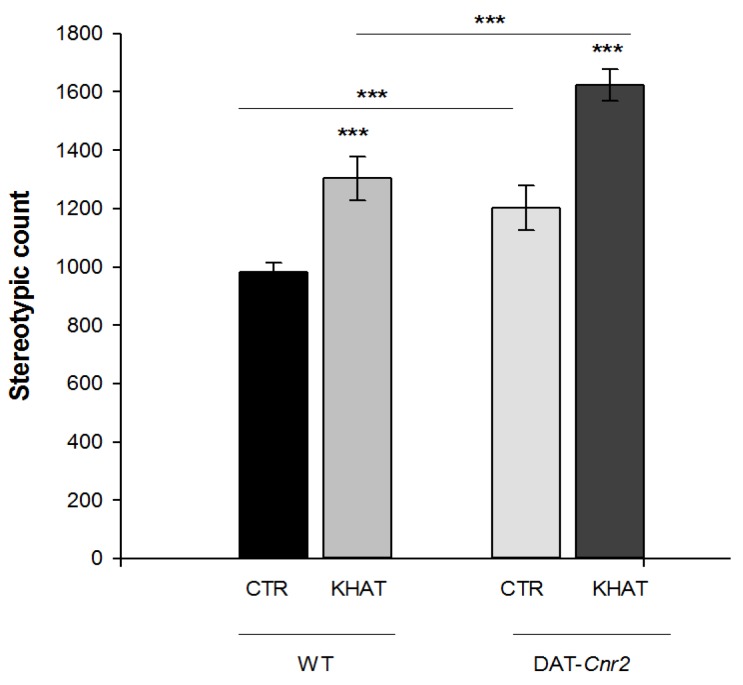
Sub-acute effect of khat extract on stereotypic count in the activity monitor apparatus in wild type (WT) and DAT-*Cnr2* mice. Values are mean ± SEM (n = 6 in each group). Statistical analysis was done using student *t*-test. *** *p* < 0.001. CTR (control group received the vehicle), KHAT (treatment group received khat extract in a dose of 300 mg/kg). CTR (control group received the vehicle), KHAT (treatment group received khat extract in a dose of 300 mg/kg). The vehicle and khat extract were administered to mice once daily for seven consecutive days.

**Figure 3 molecules-24-03164-f003:**
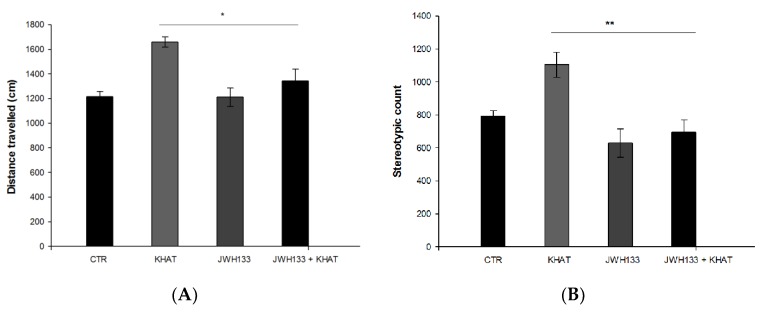
Effect of co-administration of the CB2R agonist JWH133 on khat extract induced increase in locomotor activity in wild type (WT) mice. (**A**) Effect of co-administration of the CB2R agonist JWH133 on khat extract induced increment in distance travelled (centimeters) in the activity box. (**B**) Effect of co-administration of the CB2R agonist JWH133 on khat extract induced increment in stereotypic count in the activity box. Values are mean ± SEM (n = 6 in each group). Statistical analysis was done using one-way ANOVA followed by Tukey′s multiple comparison test. ** *p* < 0.01, * *p* < 0.05. CTR (control group received the vehicle), KHAT (treatment group received khat extract in a dose of 300 mg/kg), JWH133 (treatment group received JWH133 in a dose of 5 mg/kg) and JWH133/KHAT (treatment group received khat extract 300 mg/kg in combination with JWH133 5 mg/kg). The vehicle, khat extract, and JWH133 were administered to mice once daily for seven consecutive days.

**Figure 4 molecules-24-03164-f004:**
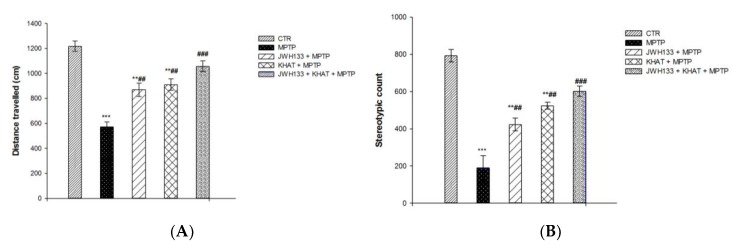
Effect of sub-acute administration of khat extract alone and in combination with JWH133 on MPTP induced motor deficits. (**A**) Effect of sub-acute administration of khat extract alone and in combination with JWH133 on distance travelled in MPTP mouse model. (**B**) Effect of sub-acute administration of khat extract alone and in combination with JWH133 on stereotypic count in MPTP mouse model. Values are mean ± SEM (n=6 in each group). Statistical analysis was done using one-way ANOVA followed by Tukey′s multiple comparison test. *** *p* <0.001, ** *p* <0.01 compared to CTR group; ^###^
*p* <0.001, ^##^
*p* <0.01 compared to MPTP lesioned group. CTR (control group received the vehicle), KHAT (treatment group received khat extract, 300 mg/kg); JWH133 (treatment group received JWH133, 5 mg/kg); JWH133 + KHAT (treatment group received JWH133 5 mg/kg and khat extract 300 mg/kg); MPTP (treatment group received 1-methyl-4-phenyl-1,2,3,6-tetrahydropyridine, MPTP, 25 mg/kg); JWH133 + MPTP (treatment group received JWH133 5 mg/kg and MPTP 25 mg/kg); KHAT + MPTP (treatment group received khat extract 300 mg/kg and MPTP 25 mg/kg); JWH133 + KHAT + MPTP (treatment group received a combination of JWH133 5 mg/kg, KHAT 300 mg/kg and MPTP 25 mg/kg). The vehicle, JWH133, khat extract and MPTP were administered to mice once daily for seven consecutive days.

**Figure 5 molecules-24-03164-f005:**
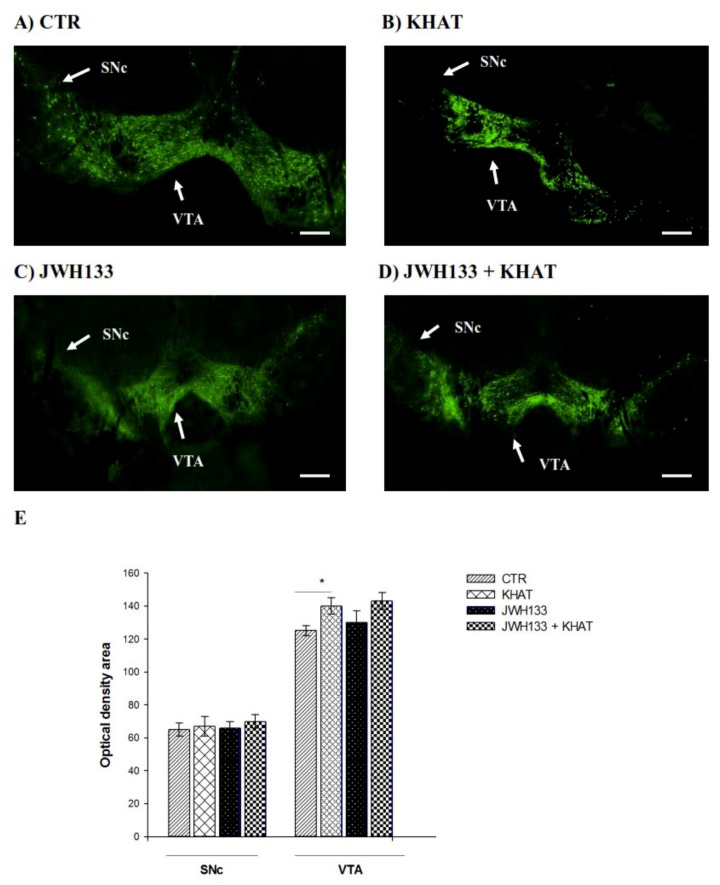
Effect of sub-acute administration of khat extract alone and in combination with JWH133 on immunohistochemical staining for TH-positive cells in WT mice. (**A**–**D**) Representative photomicrographs of TH immunoreactive neurons in the substantia nigra pars compacta (SNc) and ventral tegmental area (VTA) region. Scale bar, 100 μm. (**E**) The number TH positive cells (optical density area), and the data were expressed as mean ± SEM (n = 6 in each group). Statistical analysis was done using one-way ANOVA followed by Tukey′s multiple comparison test. * *p* < 0.05. CTR (control group received the vehicle); KHAT (treatment group received khat extract 300 mg/kg); JWH133 (treatment group received JWH133 5 mg/kg); JWH133 + KHAT (treatment group received a combination of JWH133 5 mg/kg with khat extract 300 mg/kg). The vehicle, JWH133, and khat extract were administered to mice once daily for seven consecutive days.

**Figure 6 molecules-24-03164-f006:**
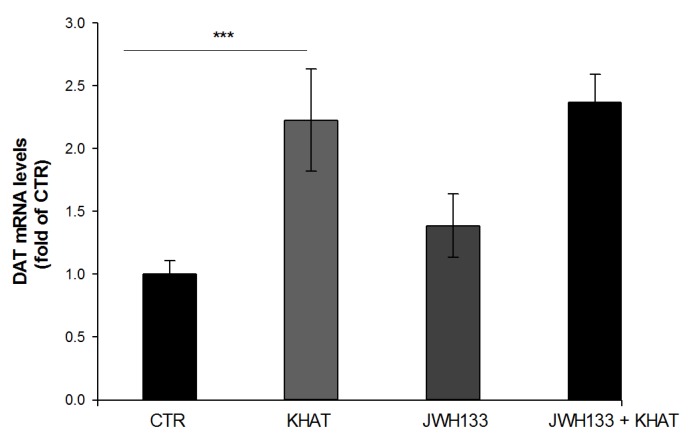
Effect of sub-acute administration of khat extract alone and in combination with JWH133 on DAT mRNA gene expression in WT mice. The quantification of dopamine transporter (DAT) mRNA levels and normalized by GAPDH mRNA in the VTA of WT mice after the administration of khat extract. Values are mean ± SEM (n=6 in each group). Statistical analysis was done using one-way ANOVA followed by Tukey′s multiple comparison test. *** *p* < 0.001. CTR (control group received the vehicle); KHAT (treatment group received khat extract 300 mg/kg); JWH133 (treatment group received JWH133 5 mg/kg); JWH133 + KHAT (treatment group received a combination of JWH133 5 mg/kg with khat extract 300 mg/kg). The vehicle, JWH133, and khat extract were administered to mice once daily for seven consecutive days.

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
