# Peer review of "Involvement of CB2 Receptors in the Neurobehavioral Effects of Catha Edulis (Vahl) Endl. (Khat) in Mice"

_molecules, 2019, doi:10.3390/molecules24173164_

Round 1

Reviewer 1 Report

Geresu et al. have addressed an interesting question: whether CB2R in dopamine (DA) neurons are involved in the hyperacivity induced by khat by studying its sub-acute effects in wild type (WT) and transgenic mice with the CB2R deleted just in DA neurons (DAT-Cnr2 null mice). They have also investigated the effects of khat in MPTP treated mice, a toxic model for Parkinson’s disease.

The research team is well acquainted with the issue investigated, and had previously contributed with many interesting findings to the field. The methods are sound and appropriate and the results are very interesting and novel. However, the pharmacologic results must be better described in the text and the effects of the selective CB2R agonist, JWH 133 (JWH), which were opposite in control compared to MPTP treated mice, should be discussed. Moreover, even though this works is focused on khat effects, the authors have disclosed an important, and biomedically relevant, effect of JWH that in their hands counteracts the motor deficits in a PD model, which strongly novel. In addition minor points should be addressed throughout the manuscript (MS).

Major comments:

Results.

-Page 3 and throughout the text, the authors should state the magnitude of the changes observed. Thus, there was an increase of around 30% (or whatever extent) in the distance travelled and stereotypic counts.

-Page 3 and Fig 1 it is not clear the hyper-locomotion of DAT-Cnr2 null mice. This should appear both in the text and be clear out of the results depicted in Fig 1. In that Fig 1 C and D it is not obvious that the animals were sub-acutely treated with khat either. Could it be possible to compare all 5 groups of animals in the same panel? Let´s say, CTR, CTR+khat, WT,   DAT-Cnr2 null mice+ vehicle and DAT-Cnr2 null mice+khat. Please consider that possibility for the sake of clarity to the readership.

-Page 3, section 2.4., Add to that sentence that JWH per se counteracted the reduction of motor activity of MPTP treated mice.

-Page 3, section 2.6. Given that optical density was scored in the TH stained sections and no actual cell counts were performed delete any reference to “TH positive cells”. The same applies to the 3rd paragraph page 5 of the Discussion. Refer to TH staining or immunoreactivity.

Discussion

-Page 4, 5th parag, the sentence “By contrast, cell type specific…stereotypic counts” is not clear. First, as mentioned above the hyperlocomotion of DAT-Cnr2 null mice is not obvious. Further, did khat further increased that locomotion? The results depicted in Fig 1 what suggests is that khat had a similar hyperlocomotive effect both in CTR and DAT-Cnr2 null mice. This should be clarified in the Discussion as well.

-Page 4, 6th parag. Include references showing that selective CB2R agonists does not affect locomotion or boosts it in a genetic model of PD.

For example:

Sánchez et al. Cancer Res. 61 (2001) 5784-5789; Yao et al. J Pharmacol Exp Ther. 2009 328:141-51; Rahn et al.  Pharmacol Biochem Behav. 2011 J98:493-502; Palomo-Garo et al. Pharmacol Res. 2016 110:181-192.

-Page 5, 2nd parag. What has been demonstrated in this work is that JWH appears to have anti-parkinsonian effect in the MPTP model, but no so a neuroprotective effect, opposite to the absence of effect in CTR mice and the blockade of khat effects. Please change the sentence to “… …that administration of JWH alone or in combination with khat showed an anti-parkinsonian and likely a neuroprotective effect…”.

-Page 5, 4th parag., the possible effect of khat on TH expression has not been studied in MPTP mice. Therefore it is an overstatement “…that khat may have a significant therapeutic effect in PD”. Reword the sentence. Indeed the present work would be more interesting if it did include TH and DAT expression studies in MPTP mice submitted to the different subacute treatments. However, this reviewer is aware of the difficulties in obtaining such data and adding them to the present MS.

What is really missing to the Discussion is a possible explanation of the fact that the subacute JWH had no effect on locomotion of control mice, is able to counteract khat hyperlocomotion in them, but shows clear motor enhancing effects in MPTP mice and does not affect the beneficial effect of khat following DA deficiency.

Finally, Conclusions should be reworded. Khat did not have antagonistic effects on that of CB2 agonist, it is the opposite. I agree that khat attenuated MPTP-induced motor deficits, as the authors mentioned, but it is uncertain that this anti-parkinsonian relies on any effect of khat on TH or DAT expression until it is proven in MPTP mice. Furthermore, the opposite effect of JWH on khat actions in control or MPTP mice casts doubts on the possible role of CB2R following DA deficiency, as already mentioned by this reviewer. Finally, the MS begins describing some of the detrimental effects of khat comsumption, thus is pertinent to compare the undesirable effects of khat on normal behavior and its possible biomedical endorsement of its treatment under disease conditions (eg PD).

Minor comments:

Abstract.

-2nd sentence please delete “…a link between this system and …” which is a repetition of 1st sentence. The absence of effect of JWH should be added to the text.

Introduction.

-Page 2, line 64 change important by relevant.

-Page 2, line 86, state that the administration was sub-acute “….following concomitant sub-acute administration…”.

-Fig. 2 A and B. The statistical difference between CTR and khat treated groups should be depicted.

-Be consistent travelled or traveled throughout the text.

Methods

-Page 6, last parag. What was the temperature while stirring? Add.

-Page 7, 1st parag, By what means the percentage yield was studied? Add.

-Page 7, 4th parag. It would be interesting to include rearing in the results. I have just scientific curiosity, since some selective CB2R agonists while not altering distance travelled or stereotypies significantly increase (2x) rears (vertical activity).

-Page 8, 3rd parag. Delete the RNA extraction of cells/well etc.

-Legends. Fig. 1, add duration and time post-injection when monitoring motor activity. Fig. 4. Indicate initial magnification of micrographs in the legend text or add a calibration barr in the actual Fig. 4. In Fig 4 E change the ordinate title “Average of TH positive cells” (no cell counts performed) by “Optical density area”.

-Fig. 5. Delete 2.33 fold since the extent of change is not indicated in any figure. This may be included in the text.

Author Response

Responses to reviewer’s comments

Reviewer 1:

Major comments

Comment: page 3 and throughout the text, the authors should state the magnitude of the changes observed. Thus, there was an increase of around 30% (or whatever extent) in the distance travelled and stereotypic counts.

Response: based on the comments from this reviewer, we stated the magnitude of the changes observed in the whole result.

Comment: page 3 and Fig 1 it is not clear the hyper-locomotion of DAT-Cnr2 null mice. This should appear both in the text and be clear out of the results depicted in Fig 1. In that Fig 1 C and D it is not obvious that the animals were sub-acutely treated with khat either. Could it be possible to compare all 5 groups of animals in the same panel? Let´s say, CTR, CTR+khat, WT,   DAT-Cnr2 null mice+ vehicle and DAT-Cnr2 null mice+khat. Please consider that possibility for the sake of clarity to the readership.

Response: in the revised manuscript, we totally changed Figure 1 and 2. Accordingly the legends of the figures were changed. We believe the new adjustment will make the figures more clear to the reader and one can easily see hyperlocomotor effect of deletion of CB2Rs per se and  their role on locomotor activity of khat.

Comment: page 3, section 2.4., Add to that sentence that JWH per se counteracted the reduction of motor activity of MPTP treated mice.

Response: according to the comment we included the role of administration of JWH133 alone on MPTP induced motor deficits in section 2.5

Comment: page 3, section 2.6. Given that optical density was scored in the TH stained sections and no actual cell counts were performed delete any reference to “TH positive cells”. The same applies to the 3rd paragraph page 5 of the Discussion. Refer to TH staining or immunoreactivity.

Response: revised according to the comments. In the mentioned sections the statement “TH positive cells” was replaced by “TH immunoreactivity”.

Comment: page 4, 5th parag, the sentence “By contrast, cell type specific…stereotypic counts” is not clear. First, as mentioned above the hyperlocomotion of DAT-Cnr2 null mice is not obvious. Further, did khat further increased that locomotion? The results depicted in Fig 1 what suggests is that khat had a similar hyperlocomotive effect both in CTR and DAT-Cnr2 null mice. This should be clarified in the Discussion as well.

Response: we believe the response mentioned above will also clarify the current comment of this reviewer. Deletion of CB2Rs per se increases locomotor activity and administration of khat extract to this mice further increases motor activity. 

Comment: page 4, 6th parag. Include references showing that selective CB2R agonists does not affect locomotion or boosts it in a genetic model of PD.

For example: Sánchez et al. Cancer Res. 61 (2001) 5784-5789; Yao et al. J Pharmacol Exp Ther. 2009 328:141-51; Rahn et al.  Pharmacol Biochem Behav. 2011 J98:493-502; Palomo-Garo et al. Pharmacol Res. 2016 110:181-192.

Response: the effect of CB2R agonists on locomotor activity is dependent on the dose of the agonist, lower dose produce hyperlocomotor activity and higher dose produce hypoactivity in the wild type mice (Chaperon and Thiébot, 1999; Fernández-Ruiz et al., 2002). In general activation of the CB2Rs by the agonists reduces motor activity in the wild type mice. But in MPTP mice model of PD, activation of the CB2Rs produce neuroprotective effect and hence increase locomotion (Chung et al., 2016; Ramirez, 2005; Shi et al., 2017; Walter et al., 2003).

Comment: page 5, 2nd parag. What has been demonstrated in this work is that JWH appears to have anti-parkinsonian effect in the MPTP model, but no so a neuroprotective effect, opposite to the absence of effect in CTR mice and the blockade of khat effects. Please change the sentence to “… …that administration of JWH alone or in combination with khat showed an anti-parkinsonian and likely a neuroprotective effect…”.

Response: we revised the statement according to the reviewer comment.

Comment: page 5, 4th parag., the possible effect of khat on TH expression has not been studied in MPTP mice. Therefore it is an overstatement “…that khat may have a significant therapeutic effect in PD”. Reword the sentence. Indeed the present work would be more interesting if it did include TH and DAT expression studies in MPTP mice submitted to the different subacute treatments. However, this reviewer is aware of the difficulties in obtaining such data and adding them to the present MS.

Response: we evaluated the locomotor effect of khat extract and JWH in MPTP treated mice. The result showed that khat alone or in combination with JWH133 reversed MPTP induced motor deficits in these mice. This shows that the compounds MIGHT have a role in PD patients since the major symptoms in PD patients are motor problems like bradykinesia and tremor. Most commonly used drugs for PD including levodopa reverse the motor deficits in these patients.  It would be very complementary to include effect of khat extract and JWH133 on TH and DAT expression but that will be our future directives. We appreciate the comment of the reviewer and reworded the statement.

Comment: What is really missing to the Discussion is a possible explanation of the fact that the subacute JWH had no effect on locomotion of control mice, is able to counteract khat hyperlocomotion in them, but shows clear motor enhancing effects in MPTP mice and does not affect the beneficial effect of khat following DA deficiency.

Response: JWH133 as a CB2R agonist produce hypolocomotion since activation of CB2Rs reduce motor activity in mice. But activation of CB2Rs produces neuroprotective effect in PD model of mice and hence reverses the motor deficit in these mice. Co-administration of JWH enhances the locomotor effect of khat in MPTP treated mice. All these concepts were included in the discussion part.   

Comment: finally, Conclusions should be reworded. Khat did not have antagonistic effects on that of CB2 agonist, it is the opposite. I agree that khat attenuated MPTP-induced motor deficits, as the authors mentioned, but it is uncertain that this anti-parkinsonian relies on any effect of khat on TH or DAT expression until it is proven in MPTP mice. Furthermore, the opposite effect of JWH on khat actions in control or MPTP mice casts doubts on the possible role of CB2R following DA deficiency, as already mentioned by this reviewer. Finally, the MS begins describing some of the detrimental effects of khat comsumption, thus is pertinent to compare the undesirable effects of khat on normal behavior and its possible biomedical endorsement of its treatment under disease conditions (eg PD).

Response: we reworded the conclusion according to the reviewer comment. Treating the motor problems in PD patients is one of the most important but not sufficient factor for a drug used to treat PD. We believe that evaluating the role of khat extract on the expression of TH and DAT will complement the motor effect of khat and to consider the compound as one therapeutic option in PD patients. JWH has an opposite effect on the role of khat in wild type mice but co-administration of JWH enhances the increase in locomotor effect after khat administration in MPTP treated mice. This is an evidence for the role of CB2Rs on the effect of khat. In the revised manuscript we included the possible adverse effects khat that may hamper its therapeutic use for chronic movement disorders.

Minor comments

Comment: 2nd sentence please delete “…a link between this system and …” which is a repetition of 1st sentence. The absence of effect of JWH should be added to the text.

Response: revised according to the comment

Comment: page 2, line 64 change important by relevant.

Response: revised according to the comment

Comment: page 2, line 86, state that the administration was sub-acute “….following concomitant sub-acute administration…”.

Response: revised according to the comment

Comment: Fig. 2 A and B. The statistical difference between CTR and khat treated groups should be depicted.

Response: this is included in the new figure 1 and 2.

Comment: be consistent travelled or traveled throughout the text.

Response: revised according to the comment

Comment: page 6, last parag. What was the temperature while stirring? Add.

Response: at room temperature and this information is added in the revised MS.

Comment: page 7, 1st parag, By what means the percentage yield was studied? Add.

Response: revised according to the comment

Comment: Page 7, 4th parag. It would be interesting to include rearing in the results. I have just scientific curiosity, since some selective CB2R agonists while not altering distance travelled or stereotypies significantly increase (2x) rears (vertical activity).

Response: we appreciate the comment of the reviewer but we believe that total distance travelled and stereotypic count are sufficient to evaluate the motor effect of a specific compound.

Comment: page 8, 3rd parag. Delete the RNA extraction of cells/well etc.

Response: revised according to the comment

Comment: legends. Fig. 1, add duration and time post-injection when monitoring motor activity.

Response: this is included in the method section

Comment: Fig. 4. Indicate initial magnification of micrographs in the legend text or add a calibration barr in the actual Fig. 4. In Fig 4 E change the ordinate title “Average of TH positive cells” (no cell counts performed) by “Optical density area”.

Response: revised according to the comment

Comment: Fig. 5. Delete 2.33 fold since the extent of change is not indicated in any figure. This may be included in the text.

Response: revised according to the comment

Reviewer 2 Report

This manuscript is clearly written, well organized and provides interesting results that may help
understanding the molecular biology of the ECS relationship with khat effects.

To provide insights into khat activity at the Endocannabinoid system, before the performance of behavioral studies, it will be interesting to know the basic pharmacology of khat and its main constituents at the cannabinoid receptors CB1 and CB2, as well as putative cannabinoid receptors such as GPR55 and GPR18. For that, binding studies and functional experiments (GTPgamma, cAMP, and or barrestin) should be provided at least at the canonical CBRs. 

In my opinion, this manuscript could be published upon addressing the aforementioned comments.

Author Response

Responses to reviewer 2’s comments

Comment: to provide insights into khat activity at the Endocannabinoid system, before the performance of behavioral studies, it will be interesting to know the basic pharmacology of khat and its main constituents at the cannabinoid receptors CB1 and CB2, as well as putative cannabinoid receptors such as GPR55 and GPR18. For that, binding studies and functional experiments (GTPgamma, cAMP, and or barrestin) should be provided at least at the canonical CBRs. 

Response: In our previous and current studies, we investigated the interaction between the ECS and Khat extract. Functional binding studies and the pharmacology of Khat on CB1 and CB2 receptors and other putative CBRs were not part of our studies and the design of these studies.  The focus was the behavioral, TH immunoblotting (data not included), IHC and DAT gene expression analysis to study the neurobehavioral effects of Khat in mice with selected deletion of CB2 cannabinoid receptors in dopamine neurons.   However, this does not mean that khat doesn’t have effect on cannabinoid receptors and this will be our future research directive since it is difficult to obtain such data and adding them to the present MS.

Reviewer 3 Report

The manuscript is a sequence of a previously published article by the same authors, in 2016, and aimed to deepen the understanding of the action of khat in the CNS of mice, and characterization of the involvement of the CBR2 endocannabinoid and dopaminergic systems in the behavioral effects induced by the plant extract. The studied was well designed, most of the results obtained are sound and give support to the author’s conclusions and to their suggestion of a potential therapeutic use of khat or khat-derived active principle(s) in the treatment of movement disorders with dopamine dysregulation.

Some points, however, need a better discussion or at least to be mentioned in the manuscript to make it acceptable for publication, as listed below.

1) In the whole paper the authors refer to khat as if it is a defined compound. Actually, the chemical composition of khat, and similar to any other plant crude extract, is that of a complex mixture, that may vary according to a number of conditions, such as plant nutrition and geographical location, time of harvest, extraction method, etc.  Besides cathinone and cathine, other active compound(s) may contribute to the observed behavioral effects. Thus, the text should make very clear that the effects seen were produced by a plant extract.

2) The khat extract used in the study was not characterized. The concentration of cathinone in the khat extract was not measured. Was the study performed with only one plant extract ? This is an obligatory information but it was not found in the methods section.

3) Khat is also known to be toxic. Besides psychoactive and addictive effects, consumption of khat has been associated to several pathologies affecting different organs, with a strong impact on pulmonary function, oral health, memory, sleep, etc. Other toxic components are present in the plant, including heavy metals. The consumption of khat was shown to counteract the effect of several medicines, including antibiotics, antihypertensive drugs, among many others, and to have teratogenic activity in rabbits. Extracts of khat were shown to inhibit detoxifying enzymes in a cathinone-independent way.

These facts should at least be mentioned in the discussion section as the toxicological profile of khat would hamper its therapeutic use for long lasting or chronic movement disorders.

4) figure 5 shows that a 2.23-fold increase in mRNA encoding DAT in the VTA region in the brains of khat-treated mice. Although the number is statistically significant, it hardly can be considered biologically relevant. Usually at least a 10-fold increase in mRNA is expected for upregulation of gene expression. Data showing an increased content of DAT protein would be  amore convincing as definite proof that expression of the transporter is upregulated.

A  list of recent references covering the above mentioned aspects of khat follows:

Al Bratty et al.
TI Determination of trace metal concentrations in different parts of the khat varieties (Catha edulis) using inductively coupled plasma-mass spectroscopy technique and their human exposure assessment
SO PHARMACOGNOSY MAGAZINE
AB Background: Khat (Catha edulis, family: Celastraceae) is a plant that is native to Africa and Arab peninsula and is used for their amphetamine-like properties. Although the use of Khat is banned in Saudi Arabia, people particularly in southern Jazan province manage to get it from Yemen, and the use is increasing steadily. Objective: Five most commonly used varieties of Khat namely Gaifi, Kofat, Jahasha, Faqarah Menjed, and Faqarah Aswad were selected for the study. Materials and Methods: Metal ion concentrations were determined using inductively coupled plasma-mass spectroscopy. Since Khat is available as one bundle consisting of three parts of the plant, metal ions in all three parts were determined separately for comparison purpose. The concentrations (mg/kg) of 20 metal ions were determined in Nwaif leaves (new and smaller in size), Gafra leaves (old and larger in size), and stem of the plant and compared with the Provisional Tolerable Weekly Intake (PTWI) and acceptable daily intake (ADI) of metal ions to study the health hazards posed by them. Results: The non-essential metal ion Strontium (Sr) was present in highest abundance in all the samples with a concentration range of 498.6 18.9u3837 52.1 mg/kg followed by Copper (215.4 12.3u3054 45.2 mg/kg), Zinc (23.17 0.4u1490 32.6 mg/kg), and Manganese (108 5.8u1357 18.6 mg/kg). Several toxic heavy metal ions including Arsenic, Lead, and Cadmium were also present in trace concentrations in many samples. Conclusion: Many metal ions were observed to be present in concentrations much higher than their PTWI and ADI which further allude to the extremely hazardous nature of Khat plants. Multicomponent variate analyses were also performed using chemometric methods to establish the possible correlation between samples.
PD JUL-SEP
PY 2019
VL 15
IS 63
BP 449
EP 458
DI 10.4103/pm.pm_658_18
ER

Lim, et al

TI Effect of 95% Ethanol Khat Extract and Cathinone on in vitro Human
Recombinant Cytochrome P450 (CYP) 2C9, CYP2D6, and CYP3A4 Activity
SO EUROPEAN JOURNAL OF DRUG METABOLISM AND PHARMACOKINETICS
AB Background and ObjectiveA significant number of people worldwide consume khat on daily basis. Long term of khat chewing has shown negative impact on several organ systems. It is likely that these people are co-administered khat preparations and conventional medication, which may lead to khat-drug interactions. This study aimed to reveal the inhibitory potencies of khat ethanol extract (KEE) and its major active ingredient (cathinone) on human cytochrome P450 (CYP) 2C9, CYP2D6, and CYP3A4 enzymes activities, which are collectively responsible for metabolizing 70-80% clinically used drugs. Methods In vitro fluorescence-based enzyme assays were developed and the CYP enzyme activities were quantified in the presence and absence of KEE and cathinone employing Vivid((R)) CYP450 Screening Kits.Results KEE inhibited human CYP2C9, CYP2D6, and CYP3A4 enzyme activities with IC50 of 42, 62, and 18g/ml. On the other hand, cathinone showed negligible inhibitory effect on these CYPs. Further experiments with KEE revealed that KEE inhibited CYP2C9 via non-competitive or mixed mode with K-i of 14.7g/ml, CYP2D6 through competitive or mixed mode with K-i of 17.6g/ml, CYP3A4 by mixed inhibition mode with K-i of 12.1g/ml. ConclusionKhat-drug interactions are possible due to administration of clinical drugs metabolized by CYP2C9/CYP2D6/CYP3A4 together with khat chewing. Further in vivo studies are required to confirm our findings and identify the causative constituents of these inhibitory effects.
PD JUN
PY 2019
VL 44
IS 3
BP 423
EP 431
DI 10.1007/s13318-018-0518-2
ER

Woldeamanuel and Geta,

TI Impact of chronic khat (Catha edulis Forsk) chewing on pulmonary function test and oxygen saturation in humans: A comparative study
SO SAGE OPEN MEDICINE
AB Background: Chronic consumption of khat affects many organ systems and leads to various health disturbances in the chewers. Few studies examined the acute effects of khat ingestion on lung function parameters. However, studies which assessed the long-term effects of khat chewing on pulmonary function parameters and oxygen saturation are lacking. Objective: The aim of this study was to assess the impact of chronic Khat chewing on pulmonary function parameters and oxygen saturation among chronic Khat chewers in Wolkite, Ethiopia. Methods: A community-based comparative cross-sectional study was conducted in Wolkite, Ethiopia from 1 June 2018 to 15 August 2018. A total of 324 participants, 162 khat chewers and 162 non-chewers were included in the study. The data were collected through face-to-face interview by trained data collectors. British Medical Research Council respiratory questionnaire was used to assess respiratory symptoms. A spirometer was used to assess various lung function parameters. Moreover, oxygen saturation of hemoglobin was measured using pulse oximeter. Data were entered into CSPro version 6.2 and analyzed using SPSS version 23. Results: This study showed statistically significant (p < 0.05) reduction in the mean values of forced vital capacity, forced expiratory volume in first second and maximum ventilation volume among khat chewers as compared to non-chewers. There was no significant difference in the mean values of other lung function parameters between the two groups. Similarly, there was no significant difference (p = 0.642) in mean oxygen saturation of hemoglobin (SaO(2)) across the two groups. Conclusion: It is evident from this study that long-term khat consumption is associated with decreased mean forced vital capacity, forced expiratory volume in first second and maximum ventilation volume. Hence, there is a need for further study to strengthen the current findings and to explore the mechanisms of khat chewing effect on lung function parameters.
PD JAN 11
PY 2019
VL 7
DI 10.1177/2050312118824616
ER

Gholami, and Bahabadi,
TI Kaurene as the major constituent of the essential oils of the narcotic plant, Khat (Catha edulis Forsk)
SO NATURAL PRODUCT RESEARCH
AB Khat (Catha edulis Forsk) is a narcotic plant which contains significant amounts of amphetamines, like alkaloids. Herein, analysis of the essential oil composition showed that Khat has useful volatile chemicals in addition to its alkaloids. Results indicated that among 35 identified constituents including mono and sesquiterpenes, the diterpene kaurene, comprises the major part of the essential oil, around 50 percent of total. Kaurene is known as a potent biological agent for the treatment of cancer patients. The presence of kaurene at high levels indicates that the essential oil of Catha edulis can potentially be more effectively exploited rather than its narcotic stimulant amphetamine-like alkaloids.
PD JAN 2
PY 2019
VL 33
IS 1
BP 126
EP 129
DI 10.1080/14786419.2018.1437424
ER

PT J
Abebe, W
TI Khat: A Substance of Growing Abuse with Adverse Drug Interaction Risks
SO JOURNAL OF THE NATIONAL MEDICAL ASSOCIATION
AB The growing global availability of the stimulant shrub, khat, has aroused widespread concern. This paper is a review of possible adverse interactions between khat and conventional drugs. Khat chewing has been shown to reduce the bioavailabilities of orally co-administered antibiotics, ampicillin, amoxicillin, cephradine and tetracycline.HCl, and the antimalarial drug, chloroquine. The cardiovascular and central nervous system (CNS) stimulant effects of monoamine oxidase inhibitors (MAOI) and amphetamine-like drugs have been described to be enhanced by khat chewing. Khat is recognized to have the ability to counteract the effects of antihypertensive, antiarrhythmic and local anesthetic drugs, and to offset the cardioprotective action of aspirin. Depending on the amount or duration of consumption, khat has been reported to variably affect the actions of general anesthetics. Khat is likely to augment the effects and/or toxicity of different drugs due to its inhibitory action on the drug metabolizing enzyme specific mechanisms have been suggested for some of the khat-drug interactions reported, the mechanisms for other interactions are less clear. Despite the above observations, the literature reviewed is associated with a number of shortcomings, suggesting the need for further research and documentation on this area of knowledge. It is recommended that, in the interim, health care providers should be more familiar with the known and suspected adverse khat-drug interactions in order to optimally serve their patients who chew khat.
PD DEC
PY 2018
VL 110
IS 6
BP 624
EP 634
DI 10.1016/j.jnma.2018.04.001
ER

Bedada et al,
TI Effects of Khat (Catha edulis) use on catalytic activities of major drug-metabolizing cytochrome P450 enzymes and implication of pharmacogenetic variations
SO SCIENTIFIC REPORTS
AB In a one-way cross-over study, we investigated the effect of Khat, a natural amphetamine-like psychostimulant plant, on catalytic activities of five major drug-metabolizing cytochrome P450 (CYP) enzymes. After a one-week Khat abstinence, 63 Ethiopian male volunteers were phenotyped using cocktail probe drugs (caffeine, losartan, dextromethorphan, omeprazole). Phenotyping was repeated after a one-week daily use of 400 g fresh Khat leaves. Genotyping for CYP1A2, CYP2C9, CYP2C19, CYP2D6, CYP3A5 were done. Urinary cathinone and phenylpropanolamine, and plasma probe drugs and metabolites concentrations were quantified using LC-MS/MS. Effect of Khat on enzyme activities was evaluated by comparing caffeine/paraxanthine (CYP1A2), losartan/losartan carboxylic acid (CYP2C9), omeprazole/5-hydroxyomeprazole (CYP2C19), dextromethorphan/dextrorphan (CYP2D6) and dextromethorphan/3-methoxymorphinan (CYP3A4) metabolic ratios (MR) before and after Khat use. Wilcoxon-matched-pair-test indicated a significant increase in median CYP2D6 MR (41%, p < 0.0001), and a marginal increase in CYP3A4 and CYP2C19 MR by Khat. Repeated measure ANOVA indicated the impact of CYP1A2 and CYP2C19 genotype on Khat-CYP enzyme interactions. The median MR increased by 35% in CYP1A2*1/*1 (p = 0.07) and by 40% in carriers of defective CYP2C19 alleles (p = 0.03). Urinary log cathinone/phenylpropanolamine ratios significantly correlated with CYP2D6 genotype (p = 0.004) and CYP2D6 MR (P = 0.025). Khat significantly inhibits CYP2D6, marginally inhibits CYP3A4, and genotype-dependently inhibit CYP2C19 and CYP1A2 enzyme activities.
PD AUG 24
PY 2018
VL 8
AR 12726
DI 10.1038/s41598-018-31191-1
ER

Abdul-Mughni, et al
TI Teratogenic effects of Khat (Catha edulis) in New Zealand rabbit
SO JOURNAL OF ADVANCED VETERINARY AND ANIMAL RESEARCH
AB Objective: The present study was carried out to evaluate morphometric and histopathological abnormalities during organogenesis in liver, kidney, brain, spinal cord, heart, Lung, digestive tract and spleen in rabbit feti in response to oral administration of Khat prepared from leaves of khat tree (Catha edulis).
  Materials and methods: The current work was carried out with apparently healthy adult New Zealand rabbits (n=27; 3 males and 24 females) weighing 2.5 +/- 0.5 Kg. The female rabbits were divided into four equal groups. Three goups (low, medium and high dose groups) were treated with Khat. The groups were given 3 mL, 6 mL and 12 mL extract/Kg bwt once daily from day 8 to 18 of gestation, respectively. The control group was given distilled water only. All females were slaughtered on day 28 of gestation. Visceral organ were subjected for histopathological examinations.
  Results: Khat was found to be associated with hepatotoxicity and nephrotoxicity in rabbits. The kidney of feti of treated dams showed subcapsular hemorrhages along with mild vacuolar degeneration of some renal tubular epithelium. Glomeruli were atrophied, and moderate degenerative changes were observed in renal tubular epithelium and hemorrhages between renal tubules. The liver of the feti showed vacuolar degeneration, necrotic hepatitis, congestion of central veins and hepatic sinusoids, pyknotic clumped nuclei, hemorrhages, edema with atrophy of some hepatocytes, and hyperplasia of Megakaryocytic cells. The Khat also harmed the brain causing hemorrhage, edema, degenerative changes, swelling and necrotic changes of some nerve cells as well as supporting cells. The spinal cord was affected showing degeneration of nerve fibers in white matter and some neurons in grey matter. The heart of treated feti showed congestion of epicardial blood vessels and diffuse degeneration of heart muscles. Lung and alimentary tract only showed congestion of blood vessels.
  Conclusion: Prenatal exposure of Khat in rabbit induces harmful effects in defferent visceral organs including liver, kidney, brain, spinal cord, spleen, intestine, heart and lung.
PD MAR
PY 2018
VL 5
IS 1
BP 25
EP 36
DI 10.5455/javar.2018.e242
ER

PT J
AU Abebe, W
TI Khat and synthetic cathinones: Emerging drugs of abuse with dental
  implications
SO ORAL SURGERY ORAL MEDICINE ORAL PATHOLOGY ORAL RADIOLOGY
AB The rising global availability of the stimulant and euphoric substances, khat and synthetic cathinones, has become a cause for concern in many countries, including the United States. Both substances are illegal in United States, although this has not deterred their use. Besides central nervous system effects, these drugs also cause sympathomimetic and orodental adverse effects, similar to those of amphetamine. Although synthetic cathinones are stronger than khat in most cases, the latter additionally contains tannins, which have astringent effects on tissues components, including those in the oral cavity. Recognizing the use prevalence and reported orodental adverse effects of khat and synthetic cathinones, dental practitioners should be more familiar with these substances to optimally treat and educate their patients abusing them. This paper reviews the pharmacology and adverse effects of khat and synthetic cathinones, along with the extent of their use in United States, with particular emphasis on dental implications.
PD FEB
PY 2018
VL 125
IS 2
BP 140
EP 146
DI 10.1016/j.oooo.2017.11.015
ER

Abdelwahab, et al

TI Khat Induced Toxicity: Role on Its Modulating Effects on Inflammation and Oxidative Stability
SO BIOMED RESEARCH INTERNATIONAL
AB Long-term khat (Catha edulis Forsk.) chewing has negative effects on human body. Khat constituents appear to be capable of disturbing the delicate equilibrium between damaging and protective mechanisms of a cell that is essential for optimal activity, thereby producing oxidative damage. Therefore, the current study was designed to understand the role of khat on cell toxicity, oxidative stability, and inflammation. Khat was extracted using 60% methanol and assessed calorimetrically for its phenolic and flavonoid contents. 1,1-diphenyl-2-picrylhydrazyl (DPPH) radical scavenging, oxygen radical absorbance capacity (ORAC), and ferric reducing/antioxidant power (FRAP) assays were used to assess the antioxidant properties. Lipopolysaccharide and interferon gamma induced murine monocytic macrophages cell line (RAW 264.7) were used to assess khat effects on cellular inflammation, oxidative stability, and viability. Khat possesses high content of polyphenols and flavonoids. The results showed a strong potency of antioxidants in DPPH, ORAC, and FRAP assays. Khat decreases the production of the proinflammatory nitric oxide and induces cytotoxicity and reactive oxygen species inhibition. Heavy khat consumption induced-toxicity and symptoms are probably due the harmful effects of its polyphenolic contents.
PY 2018
AR 5896041
DI 10.1155/2018/5896041
ER

Sallam, et al

TI The Physiological and Ergogenic Effects of Khat (Catha edulis Forsk) Extract
SO SUBSTANCE USE & MISUSE
AB Background: Khat (Catha edulis Forsk) is a natural psychoactive substance which contains two main addictive substances; Cathine and Cathinone. Khat is widely used in east Africa and southern Arabian Peninsula. Khat chewers believe that it improves work capacity and increases energy level and alertness. That is why we aimed in this study to investigate the physiological and ergogenic effects of khat extract. Methods: This study is an experimental study conducted at the Substance Abuse Research Centre in Jazan University, Saudi Arabia. Thirty healthy young volunteers were randomly assigned into two experimental groups. The first group ingested 45 g of grounded khat leaves extract mixed with juice in the first session then placebo (juice only) in the second session. While the second group ingested the placebo in the first session and the grounded khat leaves with juice in the second session. Experiments were done between December 2012 and March 2013. We recorded the blood pressure, heart rate, grip strength, and reaction time every 15 min for 75 min after each ingestion. The study proposal was reviewed and approved by Research Ethics Committee (REC) of the Medical Research Centre in Jazan University. Results: The results showed the consumption of 45 g of grounded khat leaves contributed to the increase in blood pressure (SBP & DBP) and reaction time (p < 0.05); but had no significant effect on heart rate and grip strength (p > 0.05). Conclusions: The findings of this study showed that Khat has an acute effect on some physiological parameters. These findings support the prohibition of cathinone and cathine by the World Anti-Doping Agency (WADA, 2016).
PY 2018
VL 53
IS 1
BP 94
EP 100
DI 10.1080/10826084.2017.1325375
ER

Atlabachew, et al

TI Preparative HPLC for large scale isolation, and salting-out assisted liquid-liquid extraction based method for HPLC-DAD determination of khat (Catha edulis Forsk) alkaloids
SO CHEMISTRY CENTRAL JOURNAL
AB Background: Khat (Catha edulis Forsk) is an evergreen shrub of the Celastraceae family. It is widely cultivated in Yemen and East Africa, where its fresh leaves are habitually chewed for their momentary pleasures and stimulation as amphetamine-like effects. The main psychostimulant constituents of khat are the phenylpropylamino alkaloids: cathinone, cathine and norephedrine.
  Results: In this study, simple procedures based on preparative HPLC and salting-out assisted liquid-liquid extraction (SALLE) based methods were developed respectively for large scale isolation and the extraction of psychoactive phenylpropylamino alkaloids; cathinone, cathine and norephedrine, from khat (Catha edulis Forsk) chewing leaves, a stimulant and drug of abuse plant. The three khat alkaloids were directly isolated from the crude oxalate salt by preparative HPLC-DAD method with purity > 98%. In addition, a modified (SALLE) method has been developed and evaluated for the extraction efficiency of psychoactive phenylpropylamino alkaloids from khat (Catha edulis Forsk) chewing leaves. An in situ two steps extraction protocol was followed without dispersive SPE clean up. The method involves extraction of the samples with 1% HAc and QuEChERS salt (1.0 g of CH3COONa and 6.0 g of MgSO4) followed by subsequent in situ liquid-liquid partitioning by adding ethyl acetate and NaOH solution. The optimized method allowed recoveries of 80-86% for the three alkaloids from khat sample with relative standard deviation (RSD) values less than 15% and limits of detection (0.85-1.9 mu g/mL).
  Conclusion: The method was found to be simple, cost-effective and provides cleaner chromatogram with good selectivity and reproducibility. The SALLE based protocol provided as good results as the conventional extraction method (ultrasonic assisted extraction followed by solid phase extraction, UAE-SPE) and hence the method can be applicable in forensic and biomedical sectors.
PD OCT 17
PY 2017
VL 11
AR 107
DI 10.1186/s13065-017-0337-6
ER

Manzar, et al
TI Sleep disturbances and memory impairment among pregnant women consuming
  khat: An under-recognized problem
SO ANNALS OF THORACIC MEDICINE
AB Khat (Catha edulis) is a evergreen flowering shrub that is cultivated at high altitudes, especially in East Africa and the southwest of the Arabian Peninsula. The plant contains alkaloids, of which cathinone and cathine have structural similarity and pharmacological action similar to amphetamines. The leaves are, therefore, consumed in some regions as a psychoactive stimulant due to cultural beliefs and misperceptions on the health benefits of khat consumption. This resulted in a growing prevalence of khat consumption among pregnant women. The myriad of physiological changes associated with pregnancy impairs sleep and memory. Moreover, khat has also been shown to have adverse effects on memory and sleep. Therefore, its use during pregnancy may further aggravate those impairments. The purpose of this mini-review is to summarize the changes in sleep and memory during pregnancy and the evidence supporting a relationship between khat consumption and neurocognitive deficits and sleep dysfunctions. The misperceptions of beneficial effects of khat, the high prevalence of consumption among pregnant women, and the possibility of under-reporting of khat abuse do necessitate the development of alternative methodologies to identify cases of unreported khat abuse in pregnant women. It is proposed that screening for sleep problems and memory deficits may help identify under-reported cases of khat abuse in pregnant women.
PD OCT-DEC
PY 2017
VL 12
IS 4
BP 247
EP 251
DI 10.4103/atm.ATM_24_17
ER

Berihu, et al
TI Toxic effect of khat (Catha edulis) on memory: Systematic review and meta-analysis
SO JOURNAL OF NEUROSCIENCES IN RURAL PRACTICE
AB Background: People use khat (Catha edulis) for its pleasant stimulant effect of physical activity, consciousness, motor, and mental functions. Although there are reports assessing the effect of khat on memory, there was no study based on formal systematic review and meta-analysis. Objective: We have therefore conducted this meta-analysis to determine the level of evidence for the effect of khat (C. edulis Forsk) on memory discrepancy. Methods: MEDLINE, Cochrane Library, PubMed, Academic Search Complete, SPORTDiscus, ScienceDirect, Scopus, Web of Science, and Google Scholar were searched to retrieve the papers for this review. Keywords utilized across database search were khat, cat, chat, long-term memory, short-term memory, memory deficit, randomized control trial, and cross-sectional survey. The search was limited to studies in humans and rodents; published in English language. Result: Finding of various studies included in our meta-analysis showed that the effect of acute, and subchronic exposure to khat showed that short-term memory appears to be affected depending on the duration of exposure. However, does not have any effect on long-term memory. Conclusion: Although a number of studies regarding the current topic are limited, the evidenced showed that khat (C. edulis) induced memory discrepancy.
PD JAN-MAR
PY 2017
VL 8
IS 1
BP 30
EP 37
DI 10.4103/0976-3147.193524
ER

Getasetegn, M
TI Chemical composition of Catha edulis (khat): a review
SO PHYTOCHEMISTRY REVIEWS
AB Khat (Catha edulis) belongs to Celastraceae family which contains 60-70 genera and 850-900 species. It is an indigenous plant to Ethiopia and Yemen as the countries of origin. It is also found in many other east and southern African countries. Khat leaves are chewed by the local people for their stimulant action. The main active ingredient compounds those are responsible for this action is cathinone and a mild stimulant cathine. In addition to these khat contains several phytochemicals such as alkaloids (phenylalkylamines and cathedulins), flavonoids, steroid and triterpenoids, monoterpenes and volatile aromatic compounds, and other miscellaneous compounds like vitamins and amino acids. Hence, this paper presents a comprehensive and unified review of literatures which concerned on the phytochemical composition of khat plant. And it also provides the isolated compounds with their chemical structures.
PD OCT
PY 2016
VL 15
IS 5
BP 907
EP 920
DI 10.1007/s11101-015-9435-z
ER

Author Response

Responses to reviewer 3’s comments

Comment: in the whole paper the authors refer to khat as if it is a defined compound. Actually, the chemical composition of khat, and similar to any other plant crude extract, is that of a complex mixture, that may vary according to a number of conditions, such as plant nutrition and geographical location, time of harvest, extraction method, etc.  Besides cathinone and cathine, other active compound(s) may contribute to the observed behavioral effects. Thus, the text should make very clear that the effects seen were produced by a plant extract.

Response: we totally agree with the reviewer and replaced khat with “crude khat extract” in the whole MS.

Comment: the khat extract used in the study was not characterized. The concentration of cathinone in the khat extract was not measured. Was the study performed with only one plant extract? This is an obligatory information but it was not found in the methods section.

Response: the main active principle of khat plant is cathinone but the crude khat extract and cathinone have different pharmacologic effects, as the extract contains different compounds that may interact (synergize) to produce the observed pharmacologic effects of khat. In the current study we are interested to evaluate the effect of the crude extract only and measuring the concentration of the active principle in deed will be our future plan. The study was performed with same bath of khat extract, with same time of collection and same method of extraction hence we believe same concentration of cathinone is found in a specific dose of khat extract.  

Comment: khat is also known to be toxic. Besides psychoactive and addictive effects, consumption of khat has been associated to several pathologies affecting different organs, with a strong impact on pulmonary function, oral health, memory, sleep, etc. Other toxic components are present in the plant, including heavy metals. The consumption of khat was shown to counteract the effect of several medicines, including antibiotics, antihypertensive drugs, among many others, and to have teratogenic activity in rabbits. Extracts of khat were shown to inhibit detoxifying enzymes in a cathinone-independent way. These facts should at least be mentioned in the discussion section as the toxicological profile of khat would hamper its therapeutic use for long lasting or chronic movement disorders.

Response: in the revised manuscript we included the possible adverse effects khat that may hamper its therapeutic use for chronic movement disorders. The idea here is not to replace khat with conventional drugs but to use khat as a starting material and come up with semi-synthetic or synthethic substances. Investigating the interaction of khat with other drugs and its potential effect on hepatic cytochrome enzymes is important before considering the extract as a therapeutic option and hence future pre-clinical and clinical studies should be done.  

Comment: figure 5 shows that a 2.23-fold increase in mRNA encoding DAT in the VTA region in the brains of khat-treated mice. Although the number is statistically significant, it hardly can be considered biologically relevant. Usually at least a 10-fold increase in mRNA is expected for upregulation of gene expression. Data showing an increased content of DAT protein would be a more convincing as definite proof that expression of the transporter is upregulated.

Response:  There was enhanced TH immunoreactivity and reduced DAT gene expression in the frontal cortex and VTA of mice with selected deletion of type 2 cannabinoid receptors in dopamine neurons compared with the WT mice as we reported [23 and 78].

Reviewer 4 Report

In this study the authors investigate the role of the endocannabinoid system on the neurobehavioral effect of KHAT. The investigation of the influence of KHAT on the endocannabinoid system is relevant.

The manuscript should be improved by the following points listed here:

In figure 2/B the stereotypic count is also lower for wt mice treated with JWH133. This reduction is maybe not significant. However, for the context of the study it would be important to know the p-Value for the comparison of wt versus wt treated with JWH133. Also authors should comment this reduction in the result section. As authors have demonstrated KHAT has an effect on wt mice (Figure 1) too. This is reproduced in Figure 3. MPTP treatment is as expected reducing the traveled distance in all treatment groups. However why should the seen increase in the travel distance upon KHAT treatment be different from the KHAT treatment in wt mice? – I believe the figure is misleading here. The relative effect of KHAT treatment (respectively JWH133 treatment or the combination) on MPTP treated mice should be displayed here (relation to non-MPTP treated mice). The quality of the micrographs in Figure 4 has to be improved. While the picture is acceptable for the control, the background is much higher in the KHAT and JWH133+KHAT group. Authors call the increase in TH staining as an “increase in TH positive cells”. However they measured the mean density of labeling in the selected area. The increase in the density is rather an increase in expression than increase in positive cells (as discussed later by the authors in the discussion). The third paragraph of the discussion (It is now well accepted (…)) is redundant to the introduction. Additional information given here should be moved to the introduction.

Author Response

Responses to reviewer 4’s comments

Comment: In figure 2/B the stereotypic count is also lower for wt mice treated with JWH133. This reduction is maybe not significant. However, for the context of the study it would be important to know the p-Value for the comparison of wt versus wt treated with JWH133. Also authors should comment this reduction in the result section. As authors have demonstrated KHAT has an effect on wt mice (Figure 1) too. This is reproduced in Figure 3.

Response: Figure 2 and 3 shows the role of co-administration of JWH on khat induced motor effects in wild type and MPTP treated mice that was reproducible.

Comment: MPTP treatment is as expected reducing the traveled distance in all treatment groups. However why should the seen increase in the travel distance upon KHAT treatment be different from the KHAT treatment in wt mice? – I believe the figure is misleading here. The relative effect of KHAT treatment (respectively JWH133 treatment or the combination) on MPTP treated mice should be displayed here (relation to non-MPTP treated mice).

Response: MPTP is known to induce neurotoxicity and reduce motor effect in mice. After treatment with khat extract, khat reversed MPTP induced motor deficits in the animals. But the increase in motor activity after khat treatment in MPTP mice model is lower than the increase in motor activity after khat treatment in wild type mice since MPTP highly reduced the motor effect in the genetic model of PD. In Figure 4 (in the revised MS) we tried to show the effect khat and combination of khat and JWH on MPTP induced motor deficits in PD model of mice. 

Comment: the quality of the micrographs in Figure 4 has to be improved. While the picture is acceptable for the control, the background is much higher in the KHAT and JWH133+KHAT group.

Response: revised according to the comment of the reviewer.

Comment: authors call the increase in TH staining as an “increase in TH positive cells”. However they measured the mean density of labeling in the selected area. The increase in the density is rather an increase in expression than increase in positive cells (as discussed later by the authors in the discussion). The third paragraph of the discussion (It is now well accepted (…) is redundant to the introduction. Additional information given here should be moved to the introduction.

Response: revised according to the comment.

Round 2

Reviewer 3 Report

Most of this reviewer's concerns were addressed in the revised manuscript.

Author Response

Response to Reviewer 3's report:

Reviewer 3 acknowledges that the concerns have been addressed in the revised manuscript.

Reviewer 4 Report

Authors have not properly addressed two of my comments:

1) In figure 2/B the stereotypic count is also lower for wt mice treated with JWH133. This reduction is maybe not significant. However, for the context of the study it would be important to know the p-Value for the comparison of wt versus wt treated with JWH133. Also authors should comment this reduction in the result section.

The question is here, is this reduction wt vs we treated with JWH133 significant or not. If JWH133 is reducing the sterotypic count also in wt mice, the reduction of JWH133 on mice treated with KHAT extract is potentially not specific.

2)  As authors have demonstrated KHAT has an effect on wt mice (Figure 1) too. This is reproduced in Figure 3. MPTP treatment is as expected reducing the traveled distance in all treatment groups. However why should the seen increase in the travel distance upon KHAT treatment be different from the KHAT treatment in wt mice? – I believe the figure is misleading here. The relative effect of KHAT treatment (respectively JWH133 treatment or the combination) on MPTP treated mice should be displayed here (relation to non-MPTP treated mice).

Again as KHAT also has an influence on wt mice, the increase in travel distance upon treatment with KHAT extract for MPTP treatment is potentially unspecific and no conclusion about a specific effect of KHAT on nigrostriatal DA neurons degeneration induced by MPTP can be drawn. In addition to the graphs shown in figure 3, authors should make graphs demonstrating the relative effect (KHAT + MPTP / KHAT; MPTP / CTR; ...). The result shall be described in the result section and commented in the discussion.

Author Response

Response to comments by Reviewer 4

Comments by Reviewer 4

Comment: In figure 2 the stereotypic count is also lower for wt mice treated with JWH133. This reduction is maybe not significant. However, for the context of the study it would be important to know the p-Value for the comparison of wt versus wt treated with JWH133. Also authors should comment this reduction in the result section.

The question is here, is this reduction wt vs wt treated with JWH133 significant or not. If JWH133 is reducing the sterotypic count also in wt mice, the reduction of JWH133 on mice treated with KHAT extract is potentially not specific.

Response: Response: In Figure 2B (in the revised MS), the CB2R agonist JWH133 tended to reduce the stereotypic count compared to the control group. However, the reduction was not statistically significant (p=0.222).  On the other hand, co-administration of JWH with khat extract significantly decreased the stereotypic count compared to khat-treated mice, without showing apparent difference with vehicle-treated (control WT) mice. This is included in the result section of the re-revised MS (Page 3 lines 115-118). These data show that the agonist counteracted the increase produced by khat and brought it back to basal level, suggesting that activation of the endocannabinoid system is involved in the khat-mediated effect in this paradigm.  Considering this, it is not possible to say it is a non-specific reduction.

Comment: As authors have demonstrated KHAT has an effect on wt mice (Figure 1) too. This is reproduced in Figure 3. MPTP treatment is as expected reducing the traveled distance in all treatment groups. However why should the seen increase in the travel distance upon KHAT treatment be different from the KHAT treatment in wt mice? – I believe the figure is misleading here. The relative effect of KHAT treatment (respectively JWH133 treatment or the combination) on MPTP treated mice should be displayed here (relation to non-MPTP treated mice).

Again, as KHAT also has an influence on wt mice, the increase in travel distance upon treatment with KHAT extract for MPTP treatment is potentially unspecific and no conclusion about a specific effect of KHAT on nigrostriatal DA neurons degeneration induced by MPTP can be drawn. In addition to the graphs shown in figure 3, authors should make graphs demonstrating the relative effect (KHAT + MPTP / KHAT; MPTP / CTR; ...). The result shall be described in the result section and commented in the discussion.

Response: It is true that khat extract increased the distance travelled in the wild type mice and effectively reversed the motor deficits in MPTP treated mice. The effect observed in the wild type mice might be due to increase release of dopamine from its storage site since khat has effect on the dopaminergic pathway as evidenced by our previous studies. MPTP is a known neurotoxin that causes motor deficits through degeneration of nigrostriatal dopaminergic neurons (Chiba et al., 1985; Javitch et al., 1985; Gainetdinov et al., 1997; Bezard et al., 1999). Khat extract reversed the effect of MPTP induced motor activity by stimulating the remaining dopaminergic neurons, as MPTP does not produce a total damage of dopaminergic neurons. This is the reason why the motor performance of khat-treated MPTP rats are significantly lower than the WT as well as the khat treated WT rats. 

We have revised Figure 4 to exclude KHAT, JWH133 and JWH133+KHAT group since it is a duplicate with figure 3 (Page 18). In the revised figure, we compared khat extract, JWH133 and combination of JWH133 and khat with MPTP treated mice and with the non-treated (control) group. We believe this shows the effect of khat, JWH133 or the combination on MPTP induced motor effects. We also made a minor revision to the legend of figure 4 (Page 15, line 625-626).